# Aromatase Inhibitors and Risk of Arthritis and Carpal Tunnel Syndrome among Taiwanese Women with Breast Cancer: A Nationwide Claims Data Analysis

**DOI:** 10.3390/jcm9020566

**Published:** 2020-02-19

**Authors:** Hsu-Chih Chien, Yea-Huei Kao Yang, C. Kent Kwoh, Pavani Chalasani, Debbie L. Wilson, Wei-Hsuan Lo-Ciganic

**Affiliations:** 1Institute of Clinical Pharmacy and Pharmaceutical Sciences, College of Medicine and Health Outcome Research Center, National Cheng Kung University, Tainan 701, Taiwan; s66001028@gmail.com; 2School of Pharmacy, College of Medicine, National Cheng Kung University, Tainan 701, Taiwan; 3Division of Rheumatology, College of Medicine, University of Arizona, Tucson, AZ 85724, USA; CKwoh@arthritis.arizona.edu; 4University of Arizona Arthritis Center, University of Arizona College of Medicine, Tucson, AZ 85724, USA; 5Division of Hematology and Oncology, College of Medicine, University of Arizona, Tucson, AZ 85724, USA; pchalasani@uacc.arizona.edu; 6University of Arizona Cancer Center, Tucson, AZ 85719, USA; 7Department of Pharmaceutical Outcomes & Policy, College of Pharmacy, University of Florida, Gainesville, FL 32610, USA; debbie.wilson@ufl.edu; 8Center for Drug Evaluation and Safety, College of Pharmacy, University of Florida, Gainesville, FL 32610, USA

**Keywords:** aromatase inhibitors, arthralgia, arthritis, Asian women, breast cancer, carpal tunnel syndrome, endocrine therapy, musculoskeletal problems, Taiwanese women, taxane-based chemotherapy

## Abstract

Tamoxifen or aromatase inhibitor (AI) therapy may prevent breast cancer recurrence, however, adverse effects may lead to treatment discontinuation. Evidence regarding the occurrence of AI-associated musculoskeletal problems among Asians is scarce. We identified women with breast cancer-initiating tamoxifen or AIs from the Taiwan National Health Insurance Research Database (2007–2012). Using multivariable cause-specific hazard models, we examined the association between endocrine therapy and the risk of any arthritis and carpal tunnel syndrome, adjusting for age, prior cancer treatment, and other health status factors. Among 32,055 eligible women with breast cancer (mean age = 52.6 ± 11.5 years), 87.4% initiated tamoxifen, 3.9% initiated anastrozole, 8.0% initiated letrozole, and 0.7% initiated exemestane. AI users had a higher 1-year cumulative incidence for any arthritis (13.0% vs. 8.2%, *p* < 0.0001) and carpal tunnel syndrome (1.4% vs. 0.8%, *p* = 0.008). Compared to tamoxifen users, AI users had a higher risk of any arthritis [adjusted hazard ratio (aHR) = 1.21, 95%CI = 1.09–1.34] and carpal tunnel syndrome (aHR = 1.68, 95%CI = 1.22–2.32). No significant difference was observed in the risks of any arthritis and carpal tunnel syndrome across different AIs. Taxane use was not associated with any arthritis (aHR = 0.92, 95%CI = 0.81–1.05) or carpal tunnel syndrome (aHR = 0.97, 95%CI = 0.67–1.40) compared to other chemotherapies. Taiwanese women with breast cancer-initiating AIs had an increased risk of arthritis and carpal tunnel syndrome compared to those who initiated tamoxifen.

## 1. Introduction

Breast cancer is the most common newly diagnosed and prevalent female cancer, with 1.7 million new and 6.3 million prevalent cases worldwide in 2012 [1]. Nearly 70% of breast cancers are hormone receptor-positive. Based on stage and menopausal status, women are recommended to receive adjuvant endocrine therapy either with tamoxifen or aromatase inhibitors (AIs) to prevent recurrence [2]. Despite significantly decreased recurrence with AI therapy among post-menopausal women, AIs are associated with adverse effects that may impact patient quality of life, leading to discontinuation of treatment and an increased risk of recurrence.

Musculoskeletal problems occur commonly among AI users [3,4], with reported rates ranging widely from 5% to 74% in various studies [5,6]. AI-related musculoskeletal problems such as arthritis and carpal tunnel syndrome may cause substantial pain, impact physical activity and quality of life [5], leading to non-adherence or early discontinuation of AI therapy [7]. Risk factors of AI-associated musculoskeletal problems are not well established and published results have been inconsistent [6,7,8,9]. For example, one of three cross-sectional studies found that individuals receiving taxane-based chemotherapy were associated with a higher risk (four times higher) of having arthralgia among AI users [8], while two observational studies showed null association [7,9]. Furthermore, little is known whether the risk of musculoskeletal problems varies across different AIs.

Approximately 40% of women with newly diagnosed breast cancer are Asians [1] but the evidence regarding the occurrence of AI-associated musculoskeletal problems among Asians is scarce. The majority of the clinical trials [3,4,10] recruited a few or small proportions of Asian participants. Three small, single-group observational studies (*n* < 400) from Japan found that 35% to 72% of AI users reported joint pain [9,11,12]. Compared to women with breast cancer in Western countries, Asian women with breast cancer have different characteristics (e.g., younger-onset age) [13], responses, and tolerability to the treatment [14]. The First Adjuvant Trial on All Aromatase Inhibitors (FATA-GIM3 trial) [10] found that the use of AIs for five years was not superior to the use of tamoxifen for two years followed by three years of AI use, and the three AIs had similar efficacy in 5-year disease-free survival and overall survival. Larger studies examining the safety of AI therapy in Asian populations are needed to inform and improve patient care for individuals with breast cancer. We aimed to examine the risk of any arthritis and carpal tunnel syndrome between women with breast cancer-initiating AIs versus tamoxifen using nationwide claims data in Taiwan.

## 2. Experimental Section

### 2.1. Data Sources

Our data sources were the Taiwan National Health Insurance Research (NHIRD) and Catastrophic Illness Patient Databases (CIPD) from 2004 to 2013. NHIRD is the administrative claims data of the Taiwan National Health Insurance (NHI) Program that has covered >99% of the Taiwanese population (>23 million unique people) since 1997. NHIRD datasets include enrollment information, medical claims including inpatient, outpatient and emergency department visits, and prescription claims reimbursed by the NHI Program. The CIPD registry covers more than 30 major chronic diseases (e.g., breast cancer) and requires histological and/or clinical validations for CIPD eligibility.

### 2.2. Study Cohort and Identification of Endocrine Therapy Exposure

In this retrospective cohort study with new user design, we identified women with breast cancer-initiating endocrine therapy from 2007 to 2012. We identified endocrine therapy including tamoxifen and AIs (i.e., anastrozole, exemestane, and letrozole) using the Anatomical Therapeutic Chemical (ATC) Classification System codes [15] (Appendix A). Considering the long-term survival (>10 years on average) of breast cancer, and that early recurrence is more likely to occur within two years after treatment initiation [16], we identified new users who had no endocrine therapy within three years prior to the earliest date of endocrine therapy during the study period (i.e., index date). We excluded women with any diagnosis of other cancers, bone metastasis, or any arthritis and carpal tunnel syndrome within a year before treatment initiation (Appendix A). Women having prescription opioids, nonsteroidal anti-inflammatory drugs (NSAIDs), and/or acetaminophen for ≥90 days within a year before treatment initiation were considered to have chronic pain conditions and were excluded from the analysis. A small proportion of women with missing information on age were also excluded. We grouped women into tamoxifen versus AI groups based on their initial endocrine therapy that was captured in the NHIRD data (Figure 1).

### 2.3. Outcomes and Measures 

Two separate outcomes were examined in the 12 months after initiation of endocrine therapy: time to the first episode of (1) any arthritis including osteoarthritis, rheumatoid arthritis, and other arthritis and (2) carpal tunnel syndrome. Any arthritis and carpal tunnel syndrome were identified by at least one inpatient claim or two outpatient claims that were more than 30 days apart (Appendix A). 

### 2.4. Covariates 

Based on the prior literature on breast cancer treatment, we adjusted for a series of covariates to address potential confounding [5,6]. Covariates included age, year of endocrine therapy initiation, and history of primary tumor resection (lumpectomy or mastectomy), radiation therapy, and chemotherapy (non-taxane based or taxane-based) within a year prior to initiation of endocrine therapy. We calculated the National Cancer Institute (NCI) comorbidity index (range 0–27, higher scores indicating a higher risk of mortality in a year among cancer patients) [17]. We identified non-cancer comorbidities and/or medications within a year before treatment initiation based on diagnoses and/or medications (Appendix A). History of wrist fracture and the use of thyroxine were included as potential confounders for the outcome of carpal tunnel syndrome [18]. We also adjusted for characteristics of the hospital where patients were receiving their breast cancer treatment including locations (capital vs. non-capital) and type of hospital (medical vs. non-medical centers).

### 2.5. Statistical Analysis

Patient characteristics were summarized by endocrine therapy groups with appropriate descriptive statistics (mean and standard deviation for continuous variables, and frequency and percentage for categorical variables). We used multivariable cause-specific hazard models to estimate the risk of any arthritis and carpal tunnel syndrome within a year across different AIs compared to tamoxifen, adjusting for the covariates mentioned above such as age and history of chemotherapy within a year prior to initiation of endocrine therapy. Both unadjusted and adjusted hazard ratios (aHRs) and 95% confidence intervals (CI) were presented. Statistical significance was determined based on 2-tailed tests (*p* < 0.05).

### 2.6. Stratification/Sensitivity Analysis

Given that patients initiating different endocrine therapies or chemotherapy regimens may have different disease severity and characteristics, we conducted two additional sensitivity analyses for each outcome using propensity score (PS) matched cohorts to ensure comparability. We then estimated the risk of outcomes of interest using multivariable cause-specific hazard models. We matched the cohorts using the optimal caliper widths method with widths of 0.2 standard deviations of PS [19]. Details in variables included in each PS model and the matching algorithms were summarized in Appendix A. First, we compared the risk of any arthritis and of carpal tunnel syndrome between anastrozole versus letrozole users. We excluded exemestane users from the PS-matching analysis due to relatively small sample size as well as the use of exemestane being restricted for tamoxifen-refractory post-menopausal women in Taiwan. Second, we compared the risk of any arthritis and of carpal tunnel syndrome between women receiving taxane-based versus non-taxane-based chemotherapy. Women without any history of chemotherapy were excluded from this sensitivity analysis. All statistical analyses were performed using SAS^®^ version 9.4 (SAS Institute Inc. Cary, NC, USA).

## 3. Results

Among 32,055 eligible Taiwanese women with breast cancer (mean age: 52.6 ± 11.5 year), 87.4% initiated with tamoxifen, 3.9% with anastrozole, 8.0% with letrozole, and 0.7% with exemestane. Compared with tamoxifen users, AI users were older (60.3–61.9 vs. 51.4 years, *p* < 0.0001), more likely to have more taxane-based chemotherapy (14.8–37.8% vs. 10.8%), a higher NCI index ≥2 (12.1–13.7% vs. 5.9%), more non-cancer comorbidities such as hypertension (45.5–48.4% vs. 30.5%), and more opioid use (8.0–11.2% vs. 4.4%), as shown in Table 1. Tamoxifen users had higher proportions of receiving prior primary tumor resection than AI users (89.1% vs. 40.8–74.4%). All of these differences were statistically significant (*p* < 0.0001).

### 3.1. AI Use and Risk of Any Arthritis

As shown in Table 2, AI users had a higher incidence for any arthritis within a year after initiating treatment compared with tamoxifen users (13.0% vs. 8.2%, *p* < 0.0001). Overall, AI users were associated with a 21% higher risk of any arthritis in the year after treatment initiation (aHR =1.21, 95% CI = 1.09–1.34) compared to tamoxifen users. Among AIs, only letrozole users were associated with a statistically significant increased risk of any arthritis compared to tamoxifen users (anastrozole: aHR = 1.11, 95% CI = 0.94–1.31; letrozole: aHR = 1.27, 95% CI = 1.12–1.44; exemestane: aHR = 1.10, 95% CI = 0.74–1.63). Other factors significantly associated with an increased risk of any arthritis included older age (e.g., >75 years: aHR = 3.18, 95% CI = 2.65–3.81 compared to age <45), previously receiving radiation therapy (aHR = 1.16, 95% CI = 1.05–1.27), hypertension (aHR = 1.11, 95% CI = 1.02–1.21), dyslipidemia (aHR = 1.18, 95% CI = 1.06–1.30), affective disorders (aHR = 1.32, 95% = 1.23–1.43), opioid use (aHR = 1.37, 95% CI = 1.19–1.59), and NSAIDs/acetaminophen use (aHR = 1.32, 95% CI = 1.19–1.47) (Appendix A). Prior exposure to taxane use was not associated with the risk of any arthritis (aHR = 0.92, 95% CI = 0.81–1.05).

### 3.2. AI Use and Risk of Carpal Tunnel Syndrome

Table 2 shows that AI users had a higher incidence of carpal tunnel syndrome within a year after initiating endocrine therapy compared to tamoxifen (1.4% vs. 0.8%, *p* = 0.008). Overall, AI users were associated with a 68% higher risk of carpal tunnel syndrome in the year after treatment initiation (aHR = 1.68, 95% CI = 1.22 − 2.32) compared to tamoxifen users. Among AIs, anastrozole and letrozole users were associated with a statistically significant increased risk of carpal tunnel syndrome (anastrozole: aHR = 1.77, 95% CI = 1.07 − 2.93; letrozole: aHR = 1.65, 95% CI = 1.13 − 2.42; exemestane: aHR = 1.30, 95% CI = 0.32 − 5.30) compared to tamoxifen users. Other factors significantly associated with an increased risk of carpal tunnel syndrome included age of 45 to 54 years (aHR = 1.78, 95% CI = 1.31 − 2.44 compared to age <45 years) and having a history of affective disorders (aHR = 1.34, 95% CI = 1.06 − 1.70) (Appendix A). Prior exposure to taxane use was not associated with the risk of carpal tunnel syndrome (aHR = 0.97, 95% CI = 0.67 − 1.40) compared to the use of other chemotherapy.

### 3.3. Sensitivity/Stratification Analyses

As shown in Appendix A, no PS-matched analyses found any differences in the risk of any arthritis and carpal tunnel syndrome among women using letrozole versus anastrozole and taxane- versus non-taxane based chemotherapy.

## 4. Discussion

Our study using nationwide claims data in Taiwan yielded three key findings regarding the association between type of endocrine therapy use and the occurrence of musculoskeletal problems within a year after treatment initiation. Overall, over one-tenth of Taiwanese women with breast cancer-initiating endocrine therapy experienced any arthritis within the first year of treatment, whereas the occurrence of carpal tunnel syndrome was less than 2%. Secondly, women who used AIs were associated with a 21% and 68% increased risk of any arthritis and carpal tunnel syndrome, respectively, compared to those who used tamoxifen after adjustment for relevant confounders such as age, prior cancer treatment, and other comorbidities. Thirdly, there are no significant differences in the risk of any arthritis or carpal tunnel syndrome for women initiating letrozole versus anastrozole in the sensitivity analyses using PS-matched cohorts. Notably, prior exposure to taxane was not a risk factor for either outcome of interest in the main and sensitivity analyses.

To our knowledge, this is the first study to assess the effect of different endocrine therapies on the subsequent risk of musculoskeletal problems using a real-world, population-based claims dataset of Asian women with breast cancer. Our incidence rate of any arthritis (8% to 14%) after initiation of endocrine therapy was lower than the rates reported from the previous studies (20% to 74%) [6], while our incidence rate of carpal tunnel syndrome was similar to the existing evidence (0.1% to 2.6%) [20,21]. Potential explanations for the observed difference could result from different study designs (e.g., new users design vs. prevalent users), populations, operational definitions of the exposures and outcomes and adjustment for different confounders. For example, prior studies largely focused on self-reported arthralgia or other musculoskeletal problems among non-Asian populations. Although we applied the algorithms based on the definitions from the Centers for Medicare and Medicaid Services Chronic Condition Data Warehouse (CCW) and published literature [22], we only captured cases of any arthritis and carpal tunnel syndrome that required medical attention and were documented in the claims data. Mild symptomatic arthritis and carpal tunnel syndrome may not be captured in our study. Future validation studies are needed to evaluate the magnitudes of the underestimation of arthritis and carpal tunnel syndrome using claims data compared to patient-reported outcomes.

Consistent with prior studies [6,20,21] our study showed that AI use was associated with a moderately increased risk of any arthritis and carpal tunnel syndrome compared to tamoxifen use in Taiwanese women with breast cancer. The exact mechanisms of endocrine therapy associated with musculoskeletal problems are not fully understood. As the correlation between low estrogen levels and joint pain has been recognized [23], one potential explanation is that AIs have a stronger effect of estrogen deprivation [5] compared to tamoxifen. Despite letrozole being observed with a stronger effect of estradiol suppression than anastrozole in breast cancer tissues from women receiving letrozole or anastrozole for four months [24], our PS-matched analyses showed no significant differences between letrozole and anastrozole in the risk of any arthritis and carpal tunnel syndrome. Our findings were consistent with the results of a randomized controlled trial (with <10% Asians) [25]. No significant association between exemestane and outcomes of interest may result from a small sample size of exemestane users and a relatively low incidence rate of carpal tunnel syndrome.

Moreover, relationships between any arthritis and carpal tunnel syndrome and other factors including age, taxane use, affective disorders, and analgesic use observed in our study have noteworthy implications for future research and patient care. AI use may increase arthritis risk or worsen arthritis symptoms, especially among older adults. Prior exposure to taxane use has been considered a risk factor of musculoskeletal problems based on one small cross-sectional study [8], but this association was not observed in our retrospective cohort study using the population-based nation-wide data. Given that endocrine therapies are often initiated after taxane treatment [2], and taxane use may cause acute pain syndromes within two to three days after infusion and last for five to seven days [26], excluding women having a history of arthritis and carpal tunnel syndrome in the 12 months prior to endocrine therapy initiation in our study allowed us to evaluate the true effects of endocrine therapy on musculoskeletal problems. The affective disorders, including depressive disorder and anxiety, are known to impact the pain symptoms and the level of pain reported [27] and may be associated with medication non-adherence [28]. As the success of endocrine therapy depends on patients’ adherence and the tolerability of the medication, it is important for health care professionals to discuss the potential benefits and risks with women with breast cancer receiving endocrine therapy, and take into account their informed preferences regarding their treatment.

Strengths of the current study include use of a nationwide claims database that covers nearly all Taiwanese women with breast cancer, the ability to capture complete information on comorbidities, treatments, procedures, and medications reimbursed by the NHI, and use of sophisticated PS matching approaches to balance numerous confounding factors that may influence the risk estimates. Furthermore, we recognized death as a competing risk in our analysis (i.e., women receiving endocrine therapy had to be alive to be suffering from the treatment-related adverse events). We applied cause-specific Cox models and included the National Cancer Institute Index [17] as a factor predicting the 1-year mortality after endocrine therapy [29].

Our study has several limitations. First, our study relied on administrative claims data that lacked information on cancer stage, laboratory results, and socio-behavioral information (e.g., obesity, using granulocyte-colony stimulating factor out of pocket due to it not being under NHI’s coverage in Taiwan). However, we excluded women who had a history of bone metastasis and included prior cancer treatments, procedures and image work-ups as surrogates for cancer staging in our analyses. Second, due to the inherent limitations of an observational retrospective study using claims data, it is unknown whether the dispensed drugs were actually taken by the patients. Potential unmeasured confounders cannot be ruled out from our observational study. Third, we grouped women based on their initial endocrine therapy, but they might switch to different therapies. A previous study using NHIRD data has shown that 16% of the tamoxifen users switched to AIs, 6% of the AI users switched to tamoxifen, and less than 4% of women had multiple switches [30]. If a higher proportion of women using tamoxifen switched to AIs in our study, we may have underestimated the risk of any arthritis and carpal tunnel syndrome for AIs.

## 5. Conclusions

In conclusion, in this population-based study using nationwide claims data, Taiwanese women with breast cancer-initiating AIs had an increased risk of any arthritis and carpal tunnel syndrome compared to those who initiated tamoxifen. Prior exposure to taxane was not a risk factor of any arthritis and carpal tunnel syndrome. Our findings provide real-world evidence for clinicians to consider the possibility of AI-associated musculoskeletal problems when women initiate endocrine therapy and provide timely and proper management to avoid discontinuation of AI therapy.

## Figures and Tables

**Figure 1 jcm-09-00566-f001:**
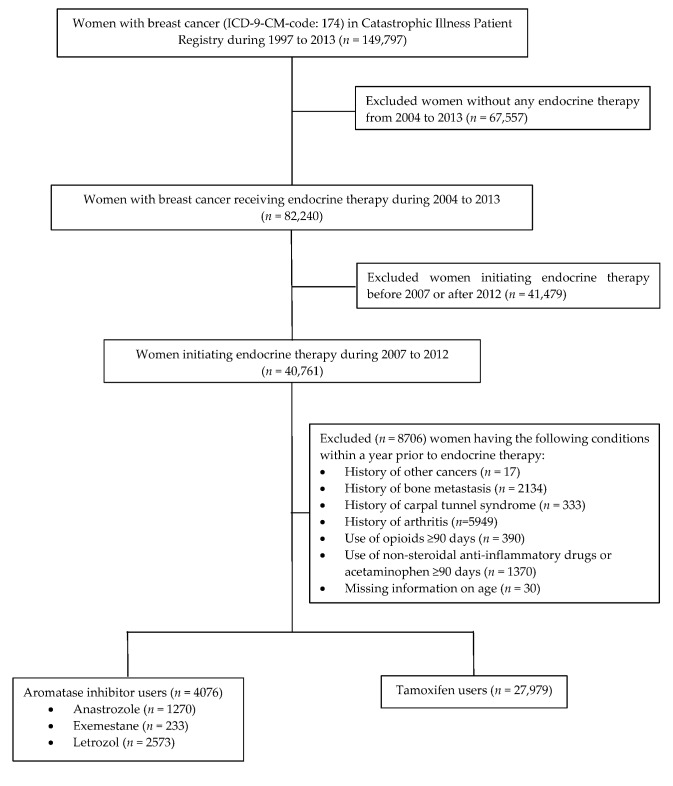
Selection of Analytical Cohort Flow Chart.

**Table 1 jcm-09-00566-t001:** Baseline Characteristics of Women with Breast Cancer by Initial Endocrine Therapy (*n*= 32,055).

Characteristics	Anastrozole (*n* = 1270, 3.9%)	Letrozole (*n* = 2573, 8.0%)	Exemestane (*n* = 233, 0.7%)	Tamoxifen (*n* = 27,979, 87.4%)	*p* Value
Mean age (SD)	61.3 (10.8)	60.3 (9.3)	61.9 (10.5)	51.4 (11.3)	<0001
Year of endocrine therapy initiation					
2007–2010	932 (4.6)	174 (0.9)	1249 (6.2)	17,825 (88.3)	<0001
2011–2012	338 (2.8)	59 (0.5)	1324 (11.1)	10,154 (85.5)	
History of treatment for breast cancer ^a^					
Primary tumor resection	883 (69.5)	1915 (74.4)	95 (40.8)	24,921 (89.1)	<0001
Radiation therapy	171 (13.5)	687 (26.7)	29 (12.5)	5386 (19.3)	<0001
Chemotherapy					
No chemotherapy	854 (67.2)	1262 (49.1)	148 (63.5)	17,711 (63.3)	<0001
Non-taxane based	228 (18.0)	339 (13.2)	31 (13.3)	7237 (25.9)	
Taxane based	188 (14.8)	972 (37.8)	54 (23.2)	3031 (10.8)	
NCI index					
0	853 (67.2)	1697 (66.0)	158 (67.8)	22,573 (80.7)	<0001
1	243 (19.1)	565 (22.0)	45 (19.3)	3753 (13.4)	
≥2	174 (13.7)	311 (12.1)	30 (12.9)	1653 (5.9)	
Comorbidities/Medications ^b^					
Hypertension	615 (48.4)	1226 (47.7)	106 (45.5)	8529 (30.5)	<0001
Diabetes	271 (21.3)	529 (20.6)	52 (22.3)	2957 (10.6)	<0001
Dyslipidemia	275 (21.7)	599 (23.3)	55 (23.6)	3953 (14.1)	<0001
Affective disorders	565 (44.5)	1225 (47.6)	104 (44.6)	11,368 (40.6)	<0001
Chronic kidney disease	24 (1.9)	43 (1.7)	4 (1.7)	264 (0.9)	0001
Liver cirrhosis	22 (1.7)	17 (0.7)	4 (1.7)	152 (0.5)	<0001
Wrist fracture	10 (0.8)	2 (0.9)	13 (0.5)	130 (0.5)	34
Thyroxine	27 (2.1)	60 (2.3)	2 (0.9)	478 (1.7)	06
Analgesic use					
Opioids	106 (8.4)	206 (8.0)	26 (11.2)	1225 (4.4)	<0001
NSAIDs/acetaminophen	1001 (78.8)	2089 (81.2)	174 (74.7)	22,653 (81.0)	<0001
Type of treatment Hospitals					
Located in capital areas	859 (67.6)	1389 (54.0)	160 (68.7)	15,565 (55.6)	<0001
Medical centers	584 (46.0)	909 (35.3)	68 (29.2)	11,852 (42.4)	<0001

Abbreviations: NCI: National Cancer Institute, NSAIDs: nonsteroidal anti-inflammatory drugs. ^a^ Prior treatment was measured within 12 months before endocrine therapy initiation. ^b^ Hypertension, diabetes, dyslipidemia and affective disorders were identified by ICD-9 codes and/or medications within 12 months before endocrine therapy initiation. The remaining were identified by ICD-9 codes.

**Table 2 jcm-09-00566-t002:** Aromatase Inhibitors Use and Risk of Any Arthritis and Carpal Tunnel Syndrome Using Cause-specific Cox model.

	Any Arthritis			Carpal Tunnel Syndrome		
Endocrine Therapy	Incidence, %	Unadjusted (HR, 95% CI)	Adjusted ^a^ (aHR, 95% CI)	Incidence, %	Unadjusted (HR, 95% CI)	Adjusteda (aHR, 95% CI)
Tamoxifen	8.2	Referent	Referent	0.8	Referent	Referent
AIs	13.0	1.66 (1.51–1.82)	1.21 (1.09–1.34)	1.4	1.52 (1.13–2.03)	1.68 (1.22–2.32)
Tamoxifen	8.2	Referent	Referent	0.8	Referent	Referent
Anastrozole	12.0	1.54 (1.31–1.81)	1.11 (0.94–1.31)	1.3	1.52 (0.93–2.48)	1.77 (1.07–2.93)
Exemestane	11.2	1.50 (1.02–2.21)	1.10 (0.74–1.63)	0.9	1.02 (0.25–4.10)	1.30 (0.32–5.30)
Letrozole	13.6	1.73 (1.55–1.94)	1.27 (1.12–1.44)	1.4	1.56 (1.10–2.21)	1.65 (1.13–2.42)

Abbreviations: aHR: adjust hazard ratio; CI: confidence interval; HR: hazard ratio ^a^ Multivariable cause-specific Cox models for the risk of any arthritis and carpal tunnel syndrome both adjusted for age, year of initiating endocrine therapy, history of primary tumor resection, radiation therapy, chemotherapy, National Cancer Institute index, history of hypertension, diabetes, dyslipidemia, affective disorders, chronic kidney disease, liver cirrhosis, use of opioids, non-steroidal anti-inflammatory drugs/acetaminophen, hospital location, and hospital type. The model for carpal tunnel syndrome additionally adjusted for a history of wrist fracture and the use of thyroxine.

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
