# Peer review of "Aromatase Inhibitors and Risk of Arthritis and Carpal Tunnel Syndrome among Taiwanese Women with Breast Cancer: A Nationwide Claims Data Analysis"

_jcm, 2020, doi:10.3390/jcm9020566_

Round 1
Reviewer 1 Report
The authors conducted a study of musculoskeletal adverse effects on a population of Taiwanese women with breast cancer taking either tamoxifen or AI as endocrine therapy. They conclude from their analysis that AI tratement increases the risks of musculoskeletal adverse effects.
In the data analyzed, there is a strong correlation of age of women and AI treatment. It is not clear in the method which model is used to deal with correlated data in the multivariable Cox model used. If increase in occurence of arthritis or CTS ir really due to AI treatment or only to age of women is thus not clearly demonstrated.
Author Response
Reviewers’ Comments:
Reviewer #1
- The authors conducted a study of musculoskeletal adverse effects on a population of Taiwanese women with breast cancer taking either tamoxifen or AI as endocrine therapy. They conclude from their analysis that AI treatment increases the risks of musculoskeletal adverse effects.
In the data analyzed, there is a strong correlation of age of women and AI treatment. It is not clear in the method which model is used to deal with correlated data in the multivariable Cox model used. If increase in occurrence of arthritis or CTS is really due to AI treatment or only to age of women is thus not clearly demonstrated.
- We have further clarified that we adjusted for potential confounders such as age and history of chemotherapy in the multivariable cause-specific Cox model in the Methods section. Although AI users were older (60.3-61.9 vs. 51.4 years, p <0.0001) compared to tamoxifen users, the observed increased risk of musculoskeletal adverse events among AI users in our study was independent of age. For example, letrozole users were slightly younger than anastrozole and exemestane users, but only letrozole users were associated with an increased risk of any arthritis.
Methods (section 2.5, lines 127-130): “We used multivariable cause-specific hazard models to estimate the risk of any arthritis and carpal tunnel syndrome within a year across different AIs compared to tamoxifen, adjusting for the covariates mentioned above such as age and history of chemotherapy within a year prior to initiation of endocrine therapy.”
Results (lines 162-167): “Overall, AI users were independently associated with a 21% higher risk of any arthritis in the year after treatment initiation (aHR =1.21, 95% CI=1.09-1.34) compared to tamoxifen users. Among AIs, only letrozole users were associated with a statistically significant increased risk of any arthritis compared to tamoxifen users (anastrozole: aHR=1.11, 95% CI=0.94-1.31; letrozole: aHR=1.27, 95% CI=1.12-1.44; exemestane: aHR=1.10, 95% CI=0.74-1.63).”

Reviewer 2 Report
interesting study, well conducted even the topic is well known
good english
line 210: "To our knowledge, this is the first study to assess the effect of different endocrine therapies on 211 the subsequent risk of musculoskeletal problems using a real-world, population-based claims dataset 212 of Asian women with breast cancer"... the weakness is the retrospectivity of the study and here must be underlienined
Author Response
Reviewers’ Comments:
Reviewer #2
- English Language and Style – reviewer checked minor edits required
- We have checked carefully and made edits throughout the manuscript to address this.
- line 210: "To our knowledge, this is the first study to assess the effect of different endocrine therapies on 211 the subsequent risk of musculoskeletal problems using a real-world, population-based claims dataset 212 of Asian women with breast cancer"... the weakness is the retrospectivity of the study and here must be underlienined
- We clarified our findings drawn from an observational retrospective study as a limitation in the Discussion section.
Discussion (lines 271-273): “Second, due to the inherent limitations of an observational retrospective study using claims data, it is unknown whether the dispensed drugs were actually taken by the patients. Potential unmeasured confounders cannot be ruled out from our observational study.”
